# Diagnostic Accuracy of Routine Laboratory Tests for COVID-19

**Joshua Davis [1,*]** and **Gina Gilderman [2]**

1    Department of Emergency Medicine, University of Kansas School of Medicine and Vituity,
     929 N. St. Francis Ave, Wichita, KS 67214, USA
2    Burrell College of Osteopathic Medicine, Las Cruces, NM 88001, USA; gina.gilderman@burrell.edu
*    Correspondence: jjvwd@udel.edu

**Abstract:** Objectives: COVID-19 has ravaged healthcare systems across the globe. Availability of and timely results for PCR testing have made diagnosis in the Emergency Department challenging. Therefore, we sought to determine if routine serum laboratory tests could be diagnostic of COVID-19. Methods: All patients tested for COVID-19 at an academic hospital in Pennsylvania between 1 March 2020–28 April 2020, were retrospectively analyzed. Results of COVID-19 PCR testing and laboratory tests were recorded. Mean difference was used to determine which tests demonstrated a significant difference, with $p < 0.01$ used, due to multiple observations. The tests that met these criteria had ROC curves and sensitivity and specificity determined. Results: Of the patients identified, 553 had had any laboratory test. All tests that showed a statistically significant mean difference were lower in COVID-19 positive patients. These included white blood cell count, platelets, absolute neutrophil count, absolute lymphocyte count, absolute eosinophil count, alkaline phosphatase, albumin, troponin T, lactic acid, D-DIMER, and procalcitonin. D-Dimer was excluded for only having four tests completed in COVID-19 positive patients. The remaining tests had a specificity of 88–96%, with a sensitivity of 5–50%. Discussion: No single serum laboratory test demonstrated sensitivity for COVID-19. Some tests might be moderately specific, but this was of limited clinical use. Future research should focus on a combination of tests to diagnose COVID-19, and healthcare systems should work to obtain rapid and accurate PCR tests to diagnose COVID-19.

**Keywords:** COVID-19; coronavirus; laboratory; serum markers; diagnostic accuracy; SARS-CoV-2



## 1. Introduction

The novel coronavirus disease 2019 (COVID-19) has ravaged and overwhelmed many healthcare systems during its initial pandemic, with over 500 million cases leading to over 6 million deaths worldwide [1]. It is caused by Severe Acute Respiratory Distress Syndrome Coronavirus 2 (SARS-CoV-2). This novel viral pathogen is associated with high rates of both infectivity [2,3]. and mortality, which has led to the need to allocate scarce healthcare resources in many settings [4].

Testing for COVID-19 is typically done via nasopharyngeal, or oral, PCR, or, more recently, antigen testing. PCR tests do not have rapid turnaround times at many facilities [5], and antigen tests are known to have limited sensitivity [6]. Even PCR tests are known to be imperfect, with sensitivities near 73–85% [7,8]. The lack of universally available rapid and accurate tests leads to a diagnostic dilemma for many clinicians, especially those in acute care, like emergency medicine, urgent care, and primary care. Incorrect guidance regarding quarantining and isolation can lead to ongoing spread of this deadly virus. Recommendations for quarantining that are over-excessive can lead to lack of compliance and social and financial burdens for patients.

Serum laboratory tests are routinely available in most acute care settings with a rapid turnaround time. If there is a single or combination of laboratory tests that could strongly suggest whether a patient had COVID-19, it could allow more accurate quarantine and

isolation recommendations. Therefore, we sought to determine the diagnostic accuracy of serum laboratory tests for COVID-19.

## 2. Methods

We conducted a retrospective review of all patients who had viral testing from 1 March 2020, to 28 April 2020, at a tertiary academic medical center in central Pennsylvania. This study was approved by the Institutional Review Board of Penn State Milton S. Hershey Medical Center.

Charts were identified using the specific order for COVID-19 testing. All patients who met this criterion and had any serum laboratory test result were included.

Availability and policies regarding COVID-19 testing at our hospital have changed often during the study period. Four different tests have been available: ARUP® Laboratories (Salt Lake City, UT, USA), Quest Diagnostics® (Secaucus, NJ, USA), Pennsylvania Department of Health (Harrisburg, PA, USA), and in-house testing at our clinical laboratory (Hershey, PA, USA). PCR testing for in house COVID-19, approved under FDA Emergency Use Authorization, was targeted against two different regions of the SARS-CoV-2 genome, ORF1ab and S gene. An RNA internal control is used to detect RT-PCR failure and/or inhibition.

Data abstracted included age and sex of patients, results of COVID-19 testing, date of testing, and results of laboratory tests. Mean difference was used to determine which tests demonstrated a significant difference, with an alpha of 0.01 selected as significant, due to multiple observations. The tests that met this criterion had receiver-operator characteristic (ROC) curves and sensitivity and specificity determined. Diagnostic accuracy was determined using standard definitions. Data was managed and statistically analyzed in Microsoft® Excel (Seattle, WA, USA).

## 3. Results

Of the 1024 patients identified who had COVID-19 testing during the study period, 553 (54%) had any laboratory testing performed. Of these, 488 (88%) were negative for COVID-19 and 65 (12%) were positive. The mean age was 54 years (SD = 22 years) and the average weight was 84 kg (SD = 28 kg). Males were 45% of the sample (248/553). Of the patients where race was provided, 77% were white (422/549), 10% Other race (58/549), 9% Black (51/549), and 3% Asian (18/549). Ten percent (59/552) were Hispanic. Among patients who tested positive for COVID-19, 45% were white (29/64), 31% Other race (20/64), 14% (9/64) Asian, and 9% (6/64) Black. Among COVID-19 positive patients, 25% were Hispanic (16/64) and 48% were male (32/66). All tests that showed a mean difference were lower in COVID-19 positive patients (Table 1). These included white blood cell count, platelets, absolute neutrophil count, absolute lymphocyte count, absolute eosinophil count, alkaline phosphatase, albumin, troponin T, lactic acid, D-DIMER, and procalcitonin. D-Dimer was excluded post hoc for only having four tests completed in COVID-19 positive patients. The remaining tests had a specificity of 88–96% with a sensitivity of 5–50% (Table 2).

**Table 1.** Mean Difference of Laboratory Tests for COVID-19.

| Laboratory Value | COVID (+) *n* | COVID (+) Mean | SD of (+) Group | COVID (−) *n* | COVID (−) Mean | SD of (−) Group | SD Both Groups | *p* Value |
|---|---|---|---|---|---|---|---|---|
| White Blood Cell Count | 65 | 6.09 | 2.54 | 487 | 10.46 | 5.92 | 3.09 | **<0.001** |
| Hemoglobin | 65 | 12.87 | 1.93 | 487 | 12.36 | 2.30 | 0.36 | 0.054 |
| Platelet | 64 | 178.47 | 68.73 | 480 | 247.26 | 106.84 | 48.64 | **<0.001** |
| Abs Neutrophil Count | 64 | 4.1 | 5.30 | 469 | 7.7 | 2.29 | 2.55 | **<0.001** |
| Abs Lymphocyte Count | 64 | 0.94 | 0.61 | 468 | 1.72 | 1.52 | 0.55 | **<0.001** |
| Abs Eosinophil Count | 64 | 0.03 | 0.06 | 468 | 0.14 | 0.19 | 0.08 | **<0.001** |
| AST | 64 | 45.59 | 46.13 | 436 | 47.49 | 115.52 | 1.34 | 0.812 |

**Table 1.** *Cont.*

| Laboratory Value | COVID (+) n | COVID (+) Mean | SD of (+) Group | COVID (−) n | COVID (−) Mean | SD of (−) Group | SD Both Groups | *p* Value |
|---|---|---|---|---|---|---|---|---|
| ALT | 64 | 33.05 | 30.69 | 443 | 43.7 | 138.15 | 7.53 | 0.162 |
| Alkaline Phosphatase | 64 | 70.94 | 30.20 | 443 | 113.8 | 93.44 | 30.31 | **<0.001** |
| Total Bilirubin | 64 | 0.49 | 0.67 | 441 | 1.39 | 7.91 | 0.64 | 0.02 |
| Albumin | 63 | 3.66 | 0.48 | 438 | 4.02 | 0.60 | 0.25 | **<0.001** |
| Lactate Dehydrogenase | 38 | 330.05 | 219.38 | 263 | 307.81 | 218.07 | 15.73 | 0.562 |
| Troponin T | 36 | 0.01 | 0.03 | 240 | 0.06 | 0.30 | 0.04 | **0.013** |
| Lactate | 43 | 1.38 | 0.64 | 299 | 1.81 | 1.08 | 0.30 | **<0.001** |
| D-DIMER | 4 | 0.43 | 0.14 | 81 | 1.57 | 2.98 | 0.81 | **0.001** |
| INR | 4 | 1.4 | 0.67 | 93 | 1.54 | 0.76 | 0.10 | 0.708 |
| Thromboplastin Time | 1 | 36 | 36.00 | 27 | 31.78 | 6.72 | 2.98 | - |
| C Reactive Protein | 37 | 5.32 | 5.57 | 306 | 5.55 | 8.12 | 0.16 | 0.824 |
| Erythrocyte Sedimentation Rate | 37 | 39.51 | 22.25 | 278 | 42.66 | 31.11 | 2.23 | 0.446 |
| Procalcitonin | 52 | 0.14 | 0.19 | 356 | 1.64 | 7.11 | 1.06 | **<0.001** |
| Ferritin | 37 | 915.49 | 1125.75 | 301 | 542.01 | 1188.64 | 264.09 | 0.065 |

COVID-19: novel coronavirus disease 2019; AST: aspartate aminotransferase; ALT: alanine transferase; INR: International Normalized Ratio

**Table 2.** Diagnostic Accuracy of Select Laboratory Tests for COVID-19.

| | Direction | Sensitivity | Specificity | Area Under Curve | Cutoff |
|---|---|---|---|---|---|
| **White Blood Cell** | **Decr.** | 26.7% | 95.8% | 82.7% | 7000 cells/hpf |
| **Platelet** | **Decr.** | 14.6% | 95.0% | 73.2% | 250,000 cells/hpf |
| **Abs Neutrophil Count** | **Decr.** | 18.1% | 95.6% | 77.2% | 6000 cellls/hpf |
| **Abs Lymphocyte Count** | **Decr.** | 13.9% | 96.0% | 76.8% | 2000 cells/hpf |
| **Abs Eosinophil Count** | **Decr.** | 23.1% | 96.1% | 94.0% | 25 cells/hpf |
| **Alkaline Phosphatase** | **Decr.** | 50.0% | 93.4% | 81.4% | 80 Units/L |
| **Albumin** | **Decr.** | 11.9% | 88.6% | 74.2% | 4.0 g/dL |
| **Troponin T** | **Decr.** | 10.9% | 95.3% | 96.0% | 0.015 ng/mL |
| **Lactate** | **Decr.** | 13.1% | 96.6% | 59.2% | 2.5 mmol/L |
| **Procalcitonin** | **Decr.** | 13.6% | 93.5% | 61.2% | 0.16 ug/mL |

COVID-19: novel coronavirus disease 2019.

## 4. Discussion

Our study reviewed the diagnostic accuracy of laboratory testing for COVID-19. Many specific findings were identified, but none of these findings were sensitive. Statistically significant findings associated with COVID-19 included leukopenia, thrombocytopenia, lymphopenia, neutropenia, eosinopenia, low alkaline phosphatase, low albumin, low troponin T, low lactic acid, and low procalcitonin. These findings were specific, but not sensitive.

Leukopenia, thrombocytopenia, and lymphopenia have previously been reported in many viral illnesses and are known to be commonly seen in COVID-19 [9] and have been shown to be negative prognostic markers [10,11]. Several mechanisms for lymphopenia and thrombocytopenia have been proposed. For lymphopenia, the following mechanisms have been proposed: hyperimmune response to IL-6 may lead to lymphocyte death, SARS-CoV-2 may directly infect T cells via ACE-2 receptors or ACE-2 independent pathways, SARS-CoV-2 may directly infect the bone marrow, or COVID-19 infection may lead to exhaustion of T cells or restrict their expansion [12]. For thrombocytopenia, the theory of bone marrow infection by SARS-CoV2 remains, but there are also theories of bone

marrow suppression for hemophagocytic lymphohistiocytosis, like reaction, autoimmune platelet destruction, or platelet consumption due to microthrombi and lung damage in a mechanism similar to that seen in disseminated intravascular coagulation [13]. Our study showed that eosinopenia is associated with COVID-19 diagnosis, which has been reported previously, but is less widely known [14,15].

Our study is the first study to suggest that low alkaline phosphatase is associated with the diagnosis of COVID-19. A prior meta-analysis has shown that elevated liver functions are not associated with diagnosis of COVID-19 at presentation [16]. Interestingly, prior studies have shown elevated liver enzymes, namely alanine aminotransferase and aspartate transaminase, to be poor prognostic markers in COVID-19 [10,11]. Acute viral hepatitis from COVID-19 has also been reported, similar to other viruses [17]. The mechanism of this viral-associated hepatitis in COVID-19 is unknown, but widely accepted theories include direct viral injury, micro-thombosis, causing ischemic hepatitis, cholestasis from systemic inflammation, and non-hepatic causes of elevation in liver enzymes (i.e., muscle damage). Hypoalbuminemia has previously been reported as a poor prognostic, but not a diagnostic, marker [18]. This has been suggested to reflect endothelial damage or pulmonary capillary leakage playing a significant role in the pathogenesis of severe COVID-19 [19].

The fact that low lactate, troponin, and procalcitonin are associated with COVID-19 is likely more reflective of ruling out alternative pathologies for COVID-19 symptoms. Many patients with COVID-19 have fever, tachycardia, and tachypnea. Thus, a normal procalcitonin and lactate may be indicative of COVID-19 in a pandemic as it makes bacterial sepsis unlikely when COVID-19 has high prevalence in the population. Similarly, chest pain is also a common complaint in patients with COVID-19. During times of high prevalence, a normal troponin may be specific to COVID-19 because it makes cardiac causes of chest pain unlikely. While classic understanding of sensitivity and specificity is that they do not vary with prevalence of disease, more recent analyses have brought this concept into question [20,21].

Unfortunately, our study did not have enough positive D-DIMER tests to evaluate this as a diagnostic marker for COVID-19. Elevated D-DIMER has been associated with COVID-19 diagnosis and prognosis, with markedly elevated levels reported, even in the absence of known confirmed thrombosis [10,22].

Given the low sensitivity of each laboratory test in isolation, they really have no clinical value in ruling out COVID-19. In resource limited settings, some of these findings may suggest COVID-19 as a diagnosis, especially in times of high prevalence of the disease. Future research should focus on identifying a combination of laboratory markers to aid in the diagnosis of COVID-19 for settings in which access to rapid direct testing is unavailable. However, it is important to note that our study was not carried out in this setting, where the prevalence of other disease processes may affect the accuracy of laboratory tests for this diagnosis (e.g., malaria and thrombocytopenia). Given the lower prevalence of disease and increased availability of PCR and antigen tests for COVID-19, there will hopefully not be a need to use surrogate laboratory markers to assist in diagnosis.

## 5. Limitations

Our study was a single site retrospective review. It has the inherent limitations of both, including the possibility of limited generalizability. We included every patient who had COVID-19 testing, thus we included patients with at least moderate pretest probability of disease. Our study occurred in the Northeast United States at an academic medical center, so populations in other settings may be different. As mentioned above, the value of some tests may vary with lower disease prevalence. Since our COVID-19 testing changed and we used PCR testing as the reference standard, the differences in the test characteristics of the different PCR tests may also affect the diagnostic accuracy of laboratory tests in our analysis.

## 6. Conclusions

Based on our data, no single serum laboratory test demonstrates sensitivity for COVID-19. Some tests may be moderately specific, but are of limited clinical use, given lower prevalence and increased availability of direct antigen and PCR testing for COVID-19. Future research should focus on a combination of tests to aid in the diagnosis of COVID-19, particularly for low resource settings without access to direct rapid COVID-19 testing. Healthcare systems should work to obtain rapid and accurate PCR tests to diagnose COVID-19, as relying on laboratory findings alone is inaccurate.

**Author Contributions:** Conceptualization, J.D.; Data curation, J.D. and G.G.; Formal analysis, J.D. and G.G.; Investigation, J.D.; Project administration, J.D.; Supervision, J.D.; Writing—original draft, J.D.; Writing—review and editing, G.G. All authors have read and agreed to the published version of the manuscript.

**Funding:** This research received no external funding.

**Institutional Review Board Statement:** This study was granted exemption by the Institutional Review Board at Penn State Milton S. Hershey Medical Center.

**Informed Consent Statement:** Not applicable.

**Data Availability Statement:** Data is available from the authors upon reasonable request.

**Conflicts of Interest:** The authors have no conflict of interest relevant to this article to disclose.

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
