# Peer review of "Diagnostic Accuracy of Routine Laboratory Tests for COVID-19"

_reports, doi:10.3390/reports5030025_

Round 1

Reviewer 1 Report

This paper gives a study on the relationship between laboratory test and COVID diagnostics. It is important, and can be published in present form.

I just have one concern. The laboratory test may be useful under resource limited conditions. The single PCR test is 20~50 RMB in China depending on CT values, and the rapid antigen test kit developed by US also has the price of less than 10 dollars for each. Can the authors give a rough estimation on the price of laboratory test?

Author Response

I agree this would be useful information. Unfortunately, it is very difficult to give a true cost/price estimate. We used several different brands of PCR tests during the study period. As you may be aware, "price" in the US health system has many different meanings. For example, the charge for one of the most basic tests of a "complete blood count" can range from $10 to $110 depending solely on what hospital it is performed at. Then, what the hospital is reimbursed depends on insurance status, and the actual cost of the test has many other factors including labor costs/cost of living and economies of scale. Any estimate at cost would thus be speculatory and have little value to readers outside of our specific health system.

Reviewer 2 Report

It is  good scientific case study  involved  Diagnostic Accuracy of Routine Laboratory Tests for COVID-19   .,  but  it  needs  minor corrections :

1- There are no  keywords ,, it must  at least  (5- 7 ).

2- in any Statistical program to analyze the results ؟ write name of  program ? or you took ((supplied )) these  data  in tables  from  medical center or  hospitals???

3- there is  on  citation  for  first  author  in  reference  No. 10  (( Davis J, Umeh U, Saba R. Treatment of SARS-CoV-2 (COVID-19): A safety perspective. World J Pharmacol. 2021;10(1):1-32. doi: 10.5497/wjp.v10.i1.1 )).'

4- the results  needs  to  more discussion  to  clarify  it .

5- I accepted paper after  minor  corrections

Author Response

Thank you.

We added keywords as follows: COVID-19, coronavirus, laboratory, serum markers, diagnostic accuracy, SARS-CoV-2

We clarified that statistics were done in Microsoft Excel.

Reference number 10 is indeed our own peer reviewed and published work, but it is cited with other peer reviewed literature in context to make points.

I am unclear what specifically the reviewer is requesting to be added in the discussion, which is already nearly half of the length of the manuscript.

Thanks again.

Reviewer 3 Report

This article is a simple retrospective analysis of the value of typical hospital laboratory testing toward a diagnostic of SARS-CoV-2 infection.  The findings that clinical laboratory testing, not of a specific SARS-CoV-2 nucleic acid or antigen testing nature, is not indicative enough one way or another to rely on for COVID diagnosis is useful information for clinicians.

It would be useful to include any kind of patient information that is possible, even if it just broad age categories, etc.  Analysis of laboratory findings broken out by age, for example, would aid in a deeper understanding of this topic.

You mention you are the first to suggest low ALP is associated with COVID.  I agree this seems to be the case, but how do you reconcile this with multiple other publications suggesting heightened levels?  A few references to this are below.

Bzeizi, K., Abdulla, M., Mohammed, N. et al. Effect of COVID-19 on liver abnormalities: a systematic review and meta‐analysis. Sci Rep 11, 10599 (2021). https://doi.org/10.1038/s41598-021-89513-9

Kumar, A., Kumar, P., Dungdung, A., Kumar Gupta, A., Anurag, A., & Kumar, A. (2020). Pattern of liver function and clinical profile in COVID-19: A cross-sectional study of 91 patients. Diabetes & metabolic syndrome, 14(6), 1951–1954. https://doi.org/10.1016/j.dsx.2020.10.001

Sharifpour A, Safanavaei S, Tabaripour R, Taghizadeh F, Nakhaei M, Abadi A, Fakhar M, Banimostafavi ES, Nazar E, Aliyali M, Abedi S, Mehravaran H, Zakariaei Z, Azadeh H. Alkaline phosphatase and score of HRCT as indicators for predicting the severity of COVID-19. Ann Med Surg (Lond). 2021 Jul;67:102519. doi: 10.1016/j.amsu.2021.102519

Author Response

Thank you. We added some demographic data in the results: The mean age was 54 years (SD=22 years) and the average weight was 84 kilograms (SD=28 kilograms). Males were 45% of the sample (248/553). Of the patients where race was provided, 77% were white (422/549), 10% Other race (58/549), 9% Black (51/549), and 3% Asian (18/549). Ten percent (59/552) were Hispanic. Among patients who tested positive for COVID-19, 45% were white (29/64), 31% Other race (20/64), 14% (9/64) Asian, and 9% (6/64) Black. Among COVID-19 positive patients, 25% were Hispanic (16/64) and 48% were male (32/66). 

The systematic review that is cited actually notes no difference in LFTs among COVID-19 patients. From the reference: "We have shown in this systematic review with meta-analysis that most of the liver enzymes and coagulation parameters in patients diagnosed with SARS-CoV-2 infection are not significantly impacted by COVID-19. We have presented an in-depth analysis of pooled mean data for liver chemistries and each of the liver function test markers in patients diagnosed with SARS-CoV-2 infection upon hospital admission. The same findings were observed with the prothrombin and activated prothrombin time. In these patients, the enzymatic liver function tests for albumin, globulin, ALT, ALP, AST, GGT and TB together with the coagulation profile were not significantly associated with COVID-19 at initial presentation."

We did add a comment on this: "A prior meta-analysis has shown that elevated liver functions are not associated with diagnosis of COVID-19 at presentation."

Thank you for your thoughtful review.